# Nuclear Fraction Proteome Analyses During rAAV Production of AAV2-Plasmid-Transfected HEK-293 Cells

**DOI:** 10.3390/ijms26136315

**Published:** 2025-06-30

**Authors:** Susanne K. Golm, Raimund Hoffrogge, Kristian M. Müller

**Affiliations:** 1Department of Biotechnology, Faculty of Technology, Bielefeld University, 33501 Bielefeld, Germany; susanne.golm@uni-bielefeld.de (S.K.G.); raimund.hoffrogge@uni-bielefeld.de (R.H.); 2Cellular and Molecular Biotechnology, Faculty of Technology, Bielefeld University, 33501 Bielefeld, Germany; 3Cell Culture Technology, Faculty of Technology, Bielefeld University, 33501 Bielefeld, Germany; 4Center for Biotechnology, Bielefeld University, 33501 Bielefeld, Germany

**Keywords:** AAV production, host cell response to virus, nanoLC-ESI-MS/MS, nuclear fraction

## Abstract

Recombinant adeno-associated virus (rAAV) is the leading vector for gene replacement therapy; however, the roles and regulation of host proteins in rAAV production remain incompletely understood. In this comparative proteomic analysis, we focused on proteins in the nucleus, the epicenter of DNA uptake, transcription, capsid assembly, and packaging. HEK-293 cells were analyzed under the following three conditions: (i) untransfected, (ii) mock-transfected with the ITR and an unrelated plasmid, and (iii) triple-transfected with rAAV2 production plasmids. Cells were harvested at 24 and 72 h post-transfection, and nuclear fractions were processed using filter-aided sample preparation (FASP) followed by nano-scale liquid chromatography–tandem mass spectrometry (nLC-Orbitrap MS/MS). Across all samples, we identified 3384 proteins, revealing significant regulatory changes associated with transfection and rAAV production. Transfection alone accounted for some of the most substantial proteomic shifts, while rAAV production induced diverse regulatory changes linked to cell cycle control, structure, and metabolism. STRING analysis of significantly regulated proteins also identified an enrichment of those associated with the Gene Ontology (GO) term ‘response to virus’. Additionally, we examined proteins with reported relation to adenoviral components. Our findings help to unravel the complexity of rAAV production, identify interesting targets for further investigation, and may contribute to improving rAAV yield.

## 1. Introduction

Adeno-associated viruses are the preferred carrier for gene replacement therapies. In 2012, the first approved gene therapy in Europe was Glybera, based on the AAV1 capsid [1]. Several approvals in, e.g., Europe and the USA followed, such as Luxturna (AAV2, 2017, [2]), Zolgensma (AAV9, 2019, [3]), Hemgenix (AAV5, 2022, [4]), Upstaza (AAV2, 2022, [5]), Roctavian (AAV5, 2023, [6]), Elevidys (AAVrh74, 2024, [7]), and Beqvez (AAVRh74var, 2024, [8]). Luxturna, Zolgensma, Upstaza, and Beqvez are produced in mammalian human embryonic kidney cells (HEK) derived from the 293 clone [9].

The triple transfection of HEK-293 serves as a classic rAAV production system. This cell line was immortalized with a fragment of the Adenovirus 5 genome, features a near-triploid karyotype with some polyploidy, and stably expresses the adenoviral proteins E1A and E1B [10]. Three required plasmids provide the basis for rAAV production: the therapeutic transgene cassette flanked by the necessary inverted terminal repeats (ITRs) of the AAV genome, as well as the non-structural AAV (Rep78, Rep68, Rep52, Rep48, AAP, and MAAP) and structural (Cap VP1, VP2, and VP3) proteins. Production has also been reported in yeast [11], Sf9 insect cells [12,13], and Chinese hamster ovary cells [14]. Furthermore, AAV capsids were produced in *E. coli* [15,16].

The production of rAAV-based therapeutics poses a number of challenges, such as extensive up- and downstream processes as well as limited yield, which results in high manufacturing costs. In this context, the question arises of how rAAV production affects the host cell and how host-cell regulation processes or modifications could be used to further increase productivity. Several transcriptomic and proteomic experiments have been carried out to investigate the impact of rAAV production on the host cells. For example, a proteomic approach was performed by Strasser et al., comparing untransfected and, for rAAV5 production, transfected FreeStyle 293-F cells. This study showed that rAAV production leads to altered expression of proteins involved in processes such as metabolism, proliferation and cell death, endocytosis, and lysosomal degradation [17]. Chung et al. examined a HEK-293-derived suspension cell line transfected for rAAV9 production with polyethylenimine (PEI) by RNA-Seq and identified several up-regulated pathways, including inflammatory and antiviral responses, as well as higher protein levels of interferon-stimulated cytokines and chemokines [18]. Another transcriptome study was conducted by Wang et al. and investigated two HEK-293 cell lines from different suppliers adapted to different media in order to identify significantly regulated metabolic pathways and examine cellular characteristics of the host cell, which may support rAAV production. They compared rAAV-producing cells with the parental cell lines and found significantly enriched and up-regulated host cell innate immune response signaling pathways (e.g., RIG-I-like receptor signaling pathway, Toll-like receptor signaling, cytosolic DNA sensing, and JAK-STAT signaling). This regulation was associated with enriched gene ontology terms for host cellular stress responses, including endoplasmic reticulum stress, autophagy, and apoptosis in virus production. In contrast, fatty acid metabolism and neutral amino acid transport were down-regulated in the late phase of viral production [19]. The impact of transfection on host cells and rAAV productivity is of great interest. Lu et al. performed a kinetic multi-omics analysis of rAAV production of transfected HEK cells [20]. Another kinetic study compared the different transfection systems of virus co-infection and multi-plasmid transfection by transcriptomic and proteomic analysis [21]. In a SWATH-MS proteomic approach, Patra et al. investigated three different transfection conditions (standard, sub-optimal, and optimal transfection) in HEKT cells [22]. The reviews of Wang et al. and Gurazada et al. provide a good overview of omics studies in this field [23,24].

We aimed to provide answers to the question of how rAAV production influences the host cell’s nuclear proteome. Since gene regulation and virus assembly are located in the nucleus [25,26], we hypothesize that limiting or supporting factors for production also reside in the nucleus and that the pre-selection for nuclear proteins helps to zoom in on important changes. We analyzed untransfected adherent HEK-293 cells and compared the nuclear proteome with that of mock-transfected cells or that of rAAV2 production-transfected cells at 24 h and 72 h post-transfection. Our label-free quantitative MS proteome analysis identified more than 3300 proteins, and we discuss the relation of selected proteins to AAV production. This study provides a valuable addition to the above-mentioned, existing proteomic analysis of the whole cell proteome approaches.

## 2. Results

The nucleus serves as a critical cellular compartment for rAAV production, as it provides the essential environment and machinery for viral DNA synthesis from transfected DNA and transcription, as well as capsid assembly and packaging of the viral DNA into the capsids. Therefore, we decoded with an nLC-ESI-MS approach the nuclear proteome of (i) untransfected (HEK sample), (ii) mock-transfected (MOCK sample), and (iii) AAV2-production-transfected (AAV sample) HEK-293 cells (Figure 1). Samples were generated in triplicate for robust statistical analysis of the results. HEK-293 cells were checked for successful transfection by fluorescence microscopy 24 h and 72 h post-transfection (ptf) based on mVenus, expressed by the plasmid pITR-mVenus, and then harvested. Nuclear fractions were generated by several steps of solubilization and precipitation, and samples were prepared for nLC-ESI MS/MS measurements by ‘filter aided sample preparation’ (FASP) based on a tryptic digestion and membrane adsorption. Raw data were analyzed for protein identification and label-free quantification with Proteome Discoverer Software (see Methods for details).

Figure 2 presents images of the HEK, MOCK, and AAV samples for both harvesting time points, 24 h and 72 h ptf. In the HEK control samples no fluorescence signal was detected, whereas in MOCK and AAV samples most cells were fluorescent, indicating successful transfection. As expected, fluorescence intensity increased from 24 h ptf to 72 h ptf. Here, we were interested in the confirmation of successful transfection. In general, fluorescence microscopy is also a powerful tool for the investigation of rAAV biology [27].

### 2.1. Protein Identification and Sample Variability

Raw nLC-ESI MS/MS data of the untransfected (HEK sample), mock-transfected (MOCK sample), and rAAV-production-transfected (AAV sample) HEK-293 cells, 24 h and 72 h post-transfection, were processed and then analyzed with Proteome Discoverer software. In total, 3384 human proteins were identified in all samples combined when filtering for proteins with at least one detected unique peptide. A principal component analysis (PCA) revealed a grouping of samples mostly based on their status of transfection and time of sampling. AAV-transfected and MOCK-transfected samples were distinct from untransfected samples by principal component 1, and samples taken at 24 h ptf were distinct from the 72 h ptf samples based on principal component 2 (Figure 3A). Additionally, distribution of identified proteins over all samples is visualized in a Venn diagram (Figure 3B). For an improved overview, 24 h and 72 h samples of each sample group (AAV, MOCK, HEK) were combined. As a reliable base for quantitative analysis, 3348 proteins are found in all sample groups altogether. Whereas one protein was only identified in AAV samples and seven in HEK samples with a confident identification. Furthermore, 21 proteins are present only in AAV and MOCK samples, five proteins in AAV and HEK samples, and two proteins in MOCK and HEK samples. In addition, the rAAV proteins Rep78, Rep52, VP1, and AAP were identified in AAV samples (proteins not included in Figure 3B).

### 2.2. Identification of Top Regulated HEK-293 Proteins

To identify significantly up- and down-regulated cellular proteins across treatments, abundance ratios (fold change) were calculated for different sample groups of the same harvest time point. Either rAAV-transfected samples or MOCK-transfected samples were compared to non-transfected HEK-293 samples. To identify significant events, proteins with at least two unique peptides, a *p*-value maximum of 0.05, and a minimum fold change of 2 were set as the threshold. The results were visualized by volcano plots for each comparison. The ten proteins with the strongest up-regulation and down-regulation are labeled with their UniProt accession number in each volcano plot (Figure 4).

### 2.3. Comparison of Top Regulated Proteins

To provide a better overview, we merged the top ten regulated proteins of both comparison time points in a Venn diagram (labeled in Figure 4 and displayed in Figure 5) and then checked for overlaps.

The comparison of the top down-regulated proteins identified three shared proteins in AAV vs. HEK and AAV vs. MOCK, namely protein XRP2 (O75695), radixin (P35241-1), and isoform 2 of DNA-directed RNA polymerases I and III subunit RPAC2 (Q9Y2S0-2). One protein was down-regulated in all sample groups, namely peripheral plasma membrane protein CASK (O14936). Furthermore, this comparison revealed five proteins down-regulated in AAV vs. HEK and MOCK vs. HEK. Interestingly, five proteins were found for AAV vs. HEK in the top rank of down-regulated proteins. The comparison of AAV vs. MOCK revealed 13 exclusive down-regulated proteins. Also, 13 down-regulated proteins were only in the MOCK vs. HEK comparison. Among the top up-regulated proteins (24 h + 72 h ptf), five proteins were found in AAV vs. HEK and AAV vs. MOCK; nine in AAV vs. HEK and MOCK vs. HEK; three in AAV vs. HEK; eleven in AAV vs. MOCK; and nine just in the MOCK vs. HEK group.

Additionally, those groups of significant proteins are listed with protein name, UniProt accession, and fold change in Table 1 for down-regulated proteins and in Table 2 for up-regulated proteins. To enable a fast search of proteins of interest, tables are sorted alphabetically for the UniProt accession numbers for each comparison (AAV vs. HEK, AAV vs. MOCK, etc.).

### 2.4. Comparison of Significantly Regulated Proteins and Respective Functional Groups

The total number of significant up- and down-regulated proteins detected in the MOCK vs. HEK samples at 24 h was higher than that of regulated proteins in AAV vs. HEK samples. This observation suggests that the transfection itself had a high impact on the regulation of cellular proteins. In AAV-producing HEK-293 cells versus non-transfected HEK-293 cells, at 24 h ptf 235 proteins and at 72 h ptf 318 proteins were significantly up-regulated, while at 24 h 374 proteins and at 72 h 212 proteins were significantly down-regulated. In MOCK-transfected samples versus non-transfected samples, at 24 h ptf 215 and at 72 h ptf 132 proteins were up-regulated, and at 24 h ptf 425 and at 72 h ptf 116 proteins were down-regulated (Figure 6).

Using the STRING online tool, we further analyzed proteins regulated in rAAV-producing cells versus HEK-293 cells at both time points (24 h and 72 h ptf). We did not exclude proteins that were also regulated in the MOCK samples, as we were interested in obtaining a picture of the whole process. The proteins were put in context using gene ontology annotations assigned by the STRING tool. In the AAV vs. HEK samples, down-regulated proteins were ascribed to the following GO terms: histone modification (30 proteins), chromatin organization (38 proteins), cellular response to stress (58 proteins), and gene expression (168 proteins). Up-regulated proteins had the following GO terms: chromosome organization (49 proteins), protein transport (56 proteins), response to stress (106 proteins), and regulation of gene expression (150 proteins), as shown in Figure 7.

### 2.5. Up-Regulated Proteins in the Context of the GO Term ‘Response to Virus’

In the GO enrichment analyses, we specifically looked at the GO term ‘response to virus’ (GO:0009615) as a subgroup of the GO term ‘response to stress’. Ten proteins with this GO assignment were identified, and an overview of the regulations of these proteins is given in Table 3.

### 2.6. rAAV-Production Specific Response in HEK-293 Cells

In the literature, we found that cyclic GMP-AMP synthase (cGAS) may play a role in rAAV production [18]. In our dataset cGAS was not detected, but two proteins associated with cGAS were up-regulated in the AAV samples. In Table 4, these proteins are listed with their calculated abundance ratios.

### 2.7. Identified Proteins with Context to Adenoviral Proteins

Finally, we checked for known linkages of identified proteins to adenoviral proteins by literature research, and we identified six candidates. These proteins are listed in Table 5 with related fold changes for each of our comparisons.

## 3. Discussion

### 3.1. Protein Identification and Sample Variability

Out of the 3384 identified proteins, proteins detected only in one dataset (Figure 3B) are discussed here separately apart from the significantly regulated proteins (Section 4.2). Uniquely identified in the AAV sample (72 h ptf) was the ‘signal transducer and activator of transcription 3′ (STAT3, UniProt P40763). As a signal transducer and transcription activator, STAT3 has multiple functions reported in the context of cytokines and the associated JAKs’ functions in cell differentiation, cell division, cell growth, and immune response [28]. STAT3 plays a role in cell cycle regulation by inducing the expression of key genes for the progression from G1 to S phase [29].

The three proteins CREB-binding protein, protein phosphatase inhibitor 2, and EF-hand domain-containing protein D1 were found only in the HEK samples, which in turn implicates that these are down-regulated upon transfection.

The CREB-binding protein (Q92793-1, gene CREBBP) is a lysine acetyltransferase and modifies key factors involved in DNA replication and interacts with multiple transcription factors and thereby plays a role in transcription activation [30]. The CREBBP acetylates Ku70, and Ku70 acetylation is increased due to DNA damage by UV radiation. Furthermore, a cytoplasmic translocation of CREB-binding protein in this process was reported [31]. Human Ku70, a nuclear protein, serves as a cytosolic DNA sensor. Upon transfection with DNA or infection with DNA viruses, Ku70 translocates from the nucleus into the cytoplasm [32]. Ku70 is also known to interact with Rep proteins and accumulates in AAV replication centers [33,34]. In this study, Ku70 is increased in AAV samples 72 h ptf. The infection and production of AAVs can induce a DNA damage response [34,35,36].

Protein phosphatase inhibitor 2 (P41236, PPP1R2) influences protein phosphatase 1 [37] and is known to regulate Aurora A/Nek2a and the PP1/Aurora A complex [38]. In the AAV production analysis of Strasser et al., aurora kinase was found to be strongly down-regulated in transfected cells and consequently postulated to lead to a reduced cell proliferation [17]. Aurora kinase A is a mitotic serine/threonine kinase that plays an important role in the regulation of cell cycle progression [39] and was not found in our nuclear samples.

The EF-hand domain-containing protein D1 (Q9BUP0, EFHD1) is known to act as a calcium sensor for mitochondrial flash activation [40] and a calcium-dependent actin–bundling activity [41]. In our experiments, transfection was performed by calcium phosphate, and this may influence protein localization to the cytoplasmic fraction.

### 3.2. Comparison of Top Regulated Proteins

In the analysis of top regulated proteins, we focused on proteins that were regulated between the AAV 24 h vs. MOCK 24 h samples (with proteins also listed in the combined 24 h AAV vs. HEK), as this reflects the peak of the virus production. Four proteins were significantly up-regulated more than 32-fold (log2(fold change) > 5). These include the following: HIC2 (Q96JB3) with a fold change of 48.1; RPAC2 (Q9Y2S0-1, gene POLR1D), also mentioned above, with a fold change of 45.2; NEXN (Nexilin, Q0ZGT2 gene gene Nexilin) with a fold change of 43.2; and LGUL (Lactoylglutathione lyase/glyoxalase I, Q04760-1, gene GLO1) with a fold change of 40.0. These proteins are discussed in the following paragraphs.

HIC2 (hypermethylated in cancer 2 protein) was originally described as a transcription factor and tumor suppressor that localizes in nuclear dots [42]. A recent report indicates that HIC2 is repressed post-transcriptionally by let-7 miRNA, and this may be reduced by DICER depletion [43]. Overexpression of HIC2 in glioblastoma cells (LN229 and U251) induced G2/M cell cycle arrest by down-regulating CDK1 [44]. HIC2 carries a conserved GLDLSKK/R motif related to the PxDLSxK/R motif in adenoviral 2 and 5 E1A proteins, which mediates binding to C-terminal binding proteins [45].

The RPAC2 subunit isoform 1 (Q9Y2S0/P0DPB6), which is shared between RNA polymerase I and III, stabilizes the enzymes’ core and is encoded by the POLR1D gene [46]. PolI transcribes ribosomal RNAs, and the transcriptome of PolIII consists of short, highly structured noncoding RNAs (ncRNAs) [47]. POLR1D expression was increased in colorectal cancer samples and implicated in cancer occurrence, progression, and resistance [48,49]. Silencing of POLR1D in lung cancer cells SK-MES-1 and H2170 inhibited proliferation. In our case, the up-regulation of RPAC2 may be induced by the increased demand for ribosomes for the synthesis of the AAV capsid proteins and, to a lesser extent, also the gene of interest.

Nexilin, also called F-actin binding protein, was analyzed in the context of cardiomyopathy [50], has been used as a gene of interest in AAV-mediated gene replacement [51], and is a seemingly unexpected find in our experiment. However, since AAV2 transduction is dependent on actin and induces remodeling of the cytoskeleton [52], a crosstalk to cellular microfilaments might also happen during production and influence related proteins. Expression studies showed that nexilin colocalized with F-actin and that an N-terminal fragment tended to aggregate in the nucleus. Further, a Z-disc association may relate to a broader stress response [53].

Lactoylglutathione lyase, also named glyoxalase I (GLO1), is an important enzyme for the detoxification of methylglyoxal (2-oxopropanal), a natural reactive byproduct of metabolism. The enzyme is up-regulated in many tumors, and inhibitors have been developed. In clinical investigations, GLO1 expression correlates negatively with cancer survival, and a predominant nuclear staining has been reported [54]. Further, a correlation of expression with key cell cycle genes has been observed, and lentiviral-mediated overexpression decreased the number of Hep3B and Huh-7 cells in the G2 phase and increased the number of cells in the S phase [55].

Three proteins are also highly up-regulated in a second interesting group: AAV vs. HEK and MOCK vs. HEK samples at 24 ptf. These proteins are sorcin (P30626-1), peflin (Q9UBV8), and annexin A11 (P50995); all three are calcium-binding proteins.

Sorcin and peflin belong to the penta-EF-hand (PEF) family [56]. The up-regulation of these three proteins is potentially induced by the transfection reagent of this study. For transfection, the calcium phosphate method was used, and in this way the calcium concentration was increased. This may induce a cellular response by up-regulating these three proteins. Peflin and the apoptosis-linked gene 2 protein (ALG-2) can form a heterodimer, which dissociates in the presence of Ca^2+^ [57]. ALG-2 interacts with Annexin 11, and the latter can interact with Sorcin [58,59,60]. This shows that all three proteins are linked. Further on, Annexin 11 is also involved in the ER-to-Golgi transport [61]. Whereas peflin was reported to be a negative regulator of ER-to-Golgi transport [62].

In the AAV vs. HEK group, the protein ZNRD2 (zinc ribbon domain containing 2, O60232), also named Sjoegren syndrome/scleroderma autoantigen 1, p27, or SSSCA1, is one of the most up-regulated proteins (>100-fold AAV vs. HEK 24 h ptf). ZNRD2 is found in many tissues, overexpressed in cancers, and regulated by Myc (see below). ZNRD2 binds zinc and proteins linked to microtubules and mitosis, possesses a nuclear exit sequence, and has been found to be centromere associated [63]. The transfection might play a major role in its detection, with some contribution from AAV production.

Interestingly, the EF-hand domain-containing protein D2 (Q96C19) is highly down-regulated in the AAV vs. HEK and in the MOCK vs. HEK groups. This protein is a calcium-binding actin-binding protein [64] and has functions in rapid cytoskeletal rearrangements [65], calcium-dependent exocytosis [66], and calcium-independent exocytosis [67]. The homologous EF-hand domain-containing protein D1 is only identified in HEK samples (discussed before). These findings suggest a role for these proteins in calcium-dependent transfection. Furthermore, a regulation of these proteins by Zn^2+^ as well as Ca^2+^ was hypothesized [68].

### 3.3. Comparison of Significantly Regulated Proteins and Respective Functional Groups

Chung et al. identified DNA/chromatin organization and DNA metabolic processes in GO enrichment analysis of triple transfected suspension HEK-293 cell RNA-Seq analysis [18]. In our data, proteins of the GO terms ‘chromatin organization’ and ‘histone modification’ show lower expression levels in AAV samples than in HEK samples. On the other hand, proteins related to the GO term ‘chromosome organization’ are up-regulated. In this context it should be mentioned that Chung et al. focused on up-regulated RNAs and mainly identified up-regulation of histone subunits 6 h and 12 h ptf. Our data acquisition started at 24 h ptf and does not cover early production time points. Interestingly, Tworig et al. increased rAAV production with a histone deacetylase (HDAC) inhibitor treatment [69].

The rAAV production triggers antiviral and innate immune responses. For example, Chung et al. and Strasser et al. identified a ‘defense response’ in cells producing rAAVs [17,18]. This GO term is included in our analysis in the GO term response to stress (discussed below in the more specific subgroup `response to virus’).

### 3.4. Up-Regulated Proteins in the Context of the GO Term ‘Response to Virus’

As we were interested in proteins affected by the viral nature of rAAV production, ten identified proteins within the GO term ‘response to virus’ (GO:0009615) assignment are discussed below (see Table 3).

#### 3.4.1. Clusterin (CLUS)

Clusterin (CLUS) is a multifunctional chaperonic glycoprotein associated with diverse cellular functions. A nucleolar localization of secretory clusterin (sCLU) was reported in response to different nucleolar stresses and the association with Cajal bodies post nucleolar stresses [70]. The functional knockdown of clusterin led to altered nuclear morphology and shrunken tubulin filaments. Nuclear morphology regulators nucleophosmin 1 (NPM1) and fibrillarin (FBL) were down-regulated in CLUS knockout cells. Kadam et al. postulated a possible role of clusterin in stabilizing these proteins [70]. NPM1 and FBL were not significantly regulated in our experiment. However, this does not exclude a role in rAAV production. Razin et al. described a virus-induced relocalization of nucleolar proteins to the nucleoplasm for further uses in viral replication compartments [71]. The nucleolar proteins nucleophosmin, fibrillarin, nucleolin (NCL), nucleolar transcription factor 1 (UBF), and nucleolar and coiled-body phosphoprotein 1 (NOLC1) were not regulated in our data. However, the nuclear protein POLR1A was down-regulated in AAV samples 24 h ptf, and treacle protein (TCOFI, isoform 4 here) was up-regulated in 24 h ptf samples of MOCK and AAV. Furthermore, Fraefel et al. observed for AAV infection that AAV DNA replication occurs in compartments, the formation of these compartments is dependent on AAV Rep, AAV ITR, and helper virus, and AAV replication compartments can associate with modified promyelocytic leukemia (PML) nuclear bodies (NBs) [72]. In adenovirus infection, Ad protein E4-orf3 binds and redistributes PML NBs to filamentous structures [73,74]. PML protein was not significantly up-regulated in our AAV and MOCK samples. Therefore, in contrast to AAV infection, nuclear organization may differ in rAAV production, also in a cell-type-specific manner.

#### 3.4.2. Deoxynucleoside Triphosphate Triphosphohydrolase SAMHD1 (SAMHD1)

SAMHD1 is a triphosphohydrolase (dNTPase), which shows restrictive functions against RNA and DNA viruses. The role of SAMDH1 in cellular defense against HIV infection is the hydrolysis of dNTPs, whereby it inhibits HIV-1 reverse transcription and, with this, HIV infection. Furthermore, SAMDH1 is important for proliferation, cell cycle regulation, apoptosis, replication fork progression, and initiating immunity and DNA damage response. Phosphorylation of SAMHD1 is used by the cell to activate dNTP hydrolysis during the G2 phase of the cell. On the other hand, dephosphorylation of SAMDH during the S phase leads to higher dNTP levels [75]. Viruses such as the Epstein–Barr virus, the human herpesvirus 6/7, the human cytomegalovirus, and Kaposi sarcoma-associated herpesvirus seem to inactivate SAMHD1 by phosphorylation through viral kinases. Some viruses are known to use the ubiquitin-proteasome system to target SAMHD1 degradation by polyubiquitinylation of SAMHD1 by influencing the CRL4 or TRIM21 E3 ubiquitin ligase. Also, SUMOylation of SAMHD1 was shown to direct a dNTP-independent antiviral mechanism [76]. These findings in cellular response to other viruses give evidence that the mechanisms may also play an important role in rAAV production. Although the overall amount of SAMHD1 seemed to be increased in our approach, one has to take into account that the multiple phosphorylations lead to a distorted picture of the real activity of this factor. In this study, SAMHD1 was significantly up-regulated 72 h ptf (10-fold compared to untransfected HEK samples). This indicates that the AAV production has an impact on the SAMHD1 amount and that this protein acts as an antiviral factor during rAAV production in HEK-293 cells. As the SAMHD1 protein amount is increased, the AAV seems not to have mechanisms like other viruses that lead to a degradation of SAMHD.

#### 3.4.3. Schlafen Family Member 11 (SLFN11)

Schlafen family member 11 (SLN11) is one of the most studied Schlafen members. It plays important roles in cancer treatment and virus restriction, e.g., in the replication of RNA viruses like HIV and the *Flavivirus* genus [77]. The degradation of SLN11 by the DNA virus cytomegalovirus was reported. This degradation is induced by the viral protein RL1, which recruits SLN11 to the CRL4 E3 ubiquitin ligase complex [78]. In our study, SLFN11 was significantly up-regulated in AAV samples at 24 h (8-fold) and 72 h (6-fold) ptf but also in MOCK samples at 24 h (4-fold) ptf. In AAV samples, proteins of DNA damage response were up-regulated, and SLFN11 is known to inhibit DNA replication in response to DNA damage and block cell death. SLFN11 is recruited to sites of DNA damage and stalled replication forks in response to replication stress sensed by DNA damage responses. SLFN11 interacts with replication protein A (RPA1) and replication helicase subunit MCM3 at replication foci and selectively blocks fork progression by chromatin opening in proximity of replication initiation sites [79,80,81]. Interestingly, Li et al. showed the lack of SLN11 did not improve AAV titers in AAV-infected cells [82].

#### 3.4.4. Histone-Lysine N-Methyltransferase SETD2 (SETD2)

SETD2 is a very interesting protein based on our expression analysis, because it is only significantly regulated in AAV samples 72 h ptf. The histone methyltransferase SETD2 specifically trimethylates ‘Lys-36’ of histone H3 (H3K36me3) and methylates STAT1. The methylation of STAT1 increases interferon-alpha (IFNα)-mediated antiviral immunity. In hepatocytes the loss of SETD2 promotes HBV infection [83]. Zhang et al. reported inhibition of interferon pathway activation by SARS-CoV-2 protein NSP9 targeting SETD2 [84]. Therefore, the inhibition of SETD2 is a potentially interesting target for improving rAAV production yield. On the other hand, methylation of histone H3Lys36 is increased by splicing [85]. In rAAV-producing cells, splicing is essential for the rAAV production, since the mRNA for most AAV proteins is generated by alternative splicing.

#### 3.4.5. 2′,5′-Phosphodiesterase 12 (PDE12)

2′,5′-Phosphodiesterase 12 (PDE12) is a mitochondrial protein and is known to remove mitochondrial RNA poly(A)tails and control mitochondrial translation [86]. PDE12 degrades the second messenger 2′,5′-oligoadenylate (2-5A) and thereby inhibits IFN-mediated antiviral defense by impeding RNase-L activation [87]. How expression of PDE12 is increased in HEK-293 cells producing rAAV needs to be investigated. Also, it would be interesting if activation of PDE12 expression before transfection would improve rAAV production.

#### 3.4.6. ATP-Dependent RNA Helicase DDX3X (DDX3X)

DDX3X has functions in transcription, mRNA maturation, export, and translation [88]. Furthermore, roles in antiviral immunity [89,90] and also for replication of several RNA viruses like hepatitis C virus [91] and human immunodeficiency virus were reported [92]. In our experiment, DDX3X was significantly up-regulated in all AAV and MOCK samples. Up-regulation in AAV samples was slightly increased compared to MOCK samples. Inhibition of DDX3X may have a positive effect on AAV production and on transgene expression in HEK-293 cells.

#### 3.4.7. Heat Shock Protein Beta-1 (HSPB1)

HSPB1 (Hsp27) inhibits cytochrome-C-mediated activation of caspase-3 and -9 and reduces apoptosome formation [93,94]. Additionally, HSPB1 increases cell survival via inhibition of Bax [95]. Interestingly, HSPB1 is more up-regulated in MOCK samples 72 h ptf than in AAV samples 72 h ptf. The mechanisms behind this observation remain unknown and need to be investigated.

#### 3.4.8. Tripartite Motif-Containing Protein 6 (TRIM6)

The E3 ubiquitin ligase TRIM6 plays an important role in the activation of the type-I interferon signaling pathway [96] and promotes replication and assembly of Ebola virus [97,98]. The potential role of TRIM6 in rAAV production is an intriguing question that could advance our understanding of rAAV biology.

#### 3.4.9. Zinc Finger CCHC Domain-Containing Protein 3 (ZCCHC3)

A further interesting candidate is the zinc finger CCHC domain-containing protein 3 (ZCCHC3). In this study, the ZCCHC3 protein is similarly up-regulated in rAAV-producing and MOCK-transfected cells 24 h post-transfection. This indicates no rAAV-specific or virus-specific cellular response. Of course, the transfection process itself is an important putative bottleneck for AAV production, next to viral defense mechanisms of the host cells. Therefore, ZCCHC3 may also be interesting for rAAV production and for general transient transfection approaches. Lian et al. identified ZCCHC3 as a positive regulator in dsRNA- and dsDNA-virus-mediated cellular antiviral immune response [99,100]. Chen et al. detected increased ZCCHC3 levels in influenza virus (RNA virus)-infected cells [101]. In this context ZCCHC3 may also be a promising target for increased rAAV particle production. ZCCHC3 is also further discussed in our next chapter.

#### 3.4.10. Heat Shock Protein HSP 90-Alpha (HSP90AA1)

The heat shock protein HSP 90 is a chaperone with a broad range of reviewed functions in apoptosis, signaling pathways, cell cycle control, protein folding, and degradation [102,103]. Interestingly, HSP90 is reported to support the internalization, virus replication, and gene expression of DNA viruses, reviewed by Lubkowska et al. [104]. HSP90AA1 is significantly up-regulated in AAV samples 72 h ptf in comparison to HEK samples. Dietmair et al. also reported up-regulation of HSP90AA1 in a multi-omics analysis of recombinant protein-producing HEK-293 cells [105]. Therefore, HSP90 may also support processes in rAAV production.

### 3.5. rAAV-Production-Specific Response in HEK-293 Cells

Transcriptomic and proteomic studies on the production of rAAV by plasmid transfection in HEK-293 cells led to the identification of induction factors of specific antiviral and inflammatory responses [17,18]. The authors discuss that cyclic GMP-AMP synthase (cGAS) may play a role in rAAV production. Chung et al. investigated the kinetic transcriptional response of suspension HEK-293 cells to rAAV production following transient polyethylenimine plasmid transfection and detected low amounts of cGAS transcripts in transfected cells. In our dataset, cGAS was not detected, presumably because cGAS is a cytosolic protein. However, in our experiment, the following two proteins associated with cGAS were up-regulated in the AAV samples:

PPP6C is the catalytic subunit of protein phosphatase 6 (PP6). The phosphatase PPP6C is constitutively associated with cGAS in unstimulated cells [106]. PPP6C negatively regulates dsDNA-induced interferon regulatory factor 3 (IRF3) activation but does not affect NF-kappa-B activation [107]. PPP6C deficiency greatly inhibits herpes simplex virus 1 (HSV-1) and vesicular stomatitis virus (VSV) replication [107]. In the context of Kaposi’s sarcoma-associated herpesvirus (KSHV), PPP6C interacts with STING, and loss of PPP6C enhances STING phosphorylation. PPP6C negatively regulates the cGAS-STING pathway by removing STING phosphorylation [107]. In our data, PPP6C was significantly up-regulated in rAAV-producing cells 72 h ptf. Interestingly, PPP6C was not up-regulated in MOCK samples at any time point. These results indicate a specific host cell response to rAAV production.

The zinc finger CCHC domain-containing protein 3 (ZCCHC3) was reported to be a co-sensor of cGAS. ZCCHC3 enhances the binding of cGAS to dsDNA by directly binding to dsDNA [99]. Additionally, this protein was reported to promote the antiviral TRIM25-mediated RIG-I activation [100,108]. Therefore, ZCCHC3 may influence transfection and rAAV production.

### 3.6. Identified Proteins with Context to Adenoviral Proteins

During rAAV production in HEK-293 cells, adenoviral proteins are expressed from the helper plasmid. Among them is the E4orf4 protein, which has toxic effects on the host cells [109]. In the literature, two human proteins are described in the context of E4orf4. Kleinberger and Shenk showed that E4orf4 interacts with the protein PP2A by a direct association with the phosphatase Bα/B55 (PPP2R2A) regulatory subunit [110]. Ben-Israel et al. showed down-regulation of Myc proto-oncogene protein (MYC) by E4orf4 expression by PPP2R2A interaction [111]. PPP2R2A is up-regulated in AAV vs. HEK 72 h ptf and in AAV vs. MOCK 24 and 72 h ptf. MYC, as an important nuclear factor, was also identified in our data, but no significant regulation was observed. Our data indicate that in rAAV production, MYC expression is only marginally affected by E4orf4. Interestingly, the PP2A subunit PPP2R1B was not up-regulated in AAV and MOCK samples in comparison to HEK samples.

The up-regulated protein DDX6 binds and re-localizes with the adenoviral protein E4 11k (product of E4orf3) in cytoplasmic p-bodies [112]. DDX6 is known to have a nucleocytoplasmic localization in human cells [113]. Increased DDX6 nuclear localization in our rAAV-producing HEK-293 cells indicates a potential influence of E4 11k on DDX6 transport.

Cullin5 was up-regulated in rAAV-producing cells, but also in MOCK-transfected cells. This protein was reported to bind adenoviral proteins E1B55K and E4orf6. The complex leads to degradation of p53, but also of the adeno-associated proteins Rep and Cap [114]. Degradation of p53 promotes cell cycle and inhibits apoptosis, which presumably has a positive effect on rAAV production, whereby degradation of Rep and Cap proteins may reduce rAAV production. In our study, p53 was down-regulated in AAV samples 72 h ptf. Furthermore, the Mre11 complex is a substrate of the E4orf6/E1B55K complex and consists of Mre11, Rad50, and NBS1. This complex detects double-strand breaks and induces p53-dependent apoptosis [115]. Interestingly, Rad50 was down-regulated in AAV samples 24 h and 72 h ptf and in MOCK samples 24 h ptf. Also, Mre11 was down-regulated in MOCK and AAV samples 24 h ptf. Our data suggest a Cullin5-initiated host cell reaction to transfection but also to rAAV production. Recently, Radukic et al. showed stabilization of AAV ITRs in E. coli plasmids in the absence of the bacterial Mre11 homolog, called SbcC, suggesting a need for rAAV-producing cells to down-regulate the Mre11/Rad50 complex for faithful genome replication [116].

## 4. Materials and Methods

### 4.1. Cell Culture of HEK-293 Cells

HEK-293 cells (DSMZ no. ACC 305, Braunschweig, Germany) were cultured in Dulbecco’s Modified Eagle Medium supplemented with 10 % (*v*/*v*) fetal calf serum and 1 % (*v*/*v*) penicillin/streptomycin (all from Sigma Aldrich, Steinheim, Germany) at 37 °C and 5 % CO_2_ in T175 culture flasks (Sarstedt, Nümbrecht, Germany).

### 4.2. Transfection of HEK-293 Cells

For transfection, 9 × 10^6^ HEK-293 cells were seeded in 150 mm cell culture dishes and incubated overnight. A total amount of 45 μg DNA per 150 mm dish was transfected. One Bijou sample container was filled with 1 mL of 2× HBS (50 mM HEPES, 1.5 mM Na_2_HPO_4_, 280 mM NaCl, pH 7.05, sterile filtered), and the second one was filled with 1 mL of CaCl_2_ (0.3 M).

Three types of samples were generated, each in triplicate: First, HEK-293 cells without transfection (HEK Sample); second, HEK-293 cells transfected with plasmids pHelper (pZMB0088), pRepCap (pZMB0618), and pITR-mVenus (pZMB0522) for rAAV2 production (AAV Sample) in a molar ratio of 1:1:1; third, HEK-293 cells transfected with the pITR-mVenus and a second non-human-coding pUC19 plasmid (pZMB0169) (MOCK Sample). For MOCK transfection, cells were transfected with the same amount of the ITR plasmid as for AAV production, while the DNA amount of pHelper and pRepCap was replaced by the noncoding plasmid. The DNA was added to the CaCl_2_ solution and was transferred dropwise to the container with 2× HBS and vortexed three times for 3 s. Immediately, 2 mL of this solution was pipetted dropwise to the cells. After 24 h and 72 h of cultivation, cells of two 150 mm cell culture plates (Sarstedt) were harvested by scraping, washed three times with PBS by resuspension and pelleting (800× *g*, 5 min), and finally pelleted by centrifugation (3000× *g*, 10 min). The cell pellets were stored at −80 °C.

### 4.3. Fluorescence Microscopy

mVenus expression was imaged with an automated fluorescence microscope (DMI 6000B, Leica Microsystem, Wetzlar, Germany) 24 h and 72 h post-transfection.

### 4.4. Protein Fractionation into Nuclear Fraction

Nuclear proteins were isolated from cell pellets using different buffers. First, the pellet was resuspended by addition of 1.25 mL LB1 buffer (50 mM HEPES-KOH pH 7.5, 140 mM NaCl, 1 mM EDTA, 10% (*v*/*v*) glycerin, 0.5% (*v*/*v*) NP-40, 0.25% (*v*/*v*) Triton X-100, pH 7.5), incubation for 10 min at 4 °C on a tube roller, and centrifugation for 5 min at 2000× *g* and 4 °C. Second, the resulting pellet was resuspended in 1.25 mL LB2 buffer (10 mM Tris-HCl, 200 mM NaCl, 1 mM EDTA, 0.5 mM EGTA, pH 8.0) and centrifuged for 5 min at 2000× *g* and 4 °C. The pellet was resuspended in 225 μL LB3 buffer (10 mM Tris-HCl, 100 mM NaCl, 1 mM EDTA, 0.5 mM EGTA, 0.1% (*v*/*v*) sodium deoxycholate, 0.5% (*v*/*v*) N-lauroylsarcosine, pH 8.0). Next, ultrasonication was performed three times for 10 s (Branson S-250A sonifier, 40% amplitude, Thermo Fisher Scientific, Darmstadt, Germany) with samples cooled on ice. The sample was supplemented with 22.5 μL of 10% Triton-X 100, vortexed, and centrifuged for 10 min (at 4 °C, 13,300× *g*). The supernatant was used for further MS-sample generation. The protein amount was quantified with a bicinchoninic acid assay (BCA, UP40840A, Interchim, Montluçon, France) according to the manufacturer’s protocol with bovine serum albumin as a reference substance measured at 560 nm with an iMark microplate absorbance reader (Bio-Rad, Feldkirchen, Germany).

### 4.5. Filter-Aided Sample Preparation (FASP) for Mass Spectrometry

The filter-aided sample preparation (FASP) method was adapted after Wiśniewski et al. [117]. Moreover, 50 µg of protein was precipitated by a 9-fold acetone volume, vortexed, stored at −20 °C overnight, and centrifuged for 10 min at 13,300× *g*. After centrifugation, the supernatant was aspirated and the pellet was dried at room temperature for 10 min. The pellet was resuspended in 200 µL UA buffer (8 mM urea, 100 mM Tris-HCl, pH 9) and added to a 30 kDa cutoff centrifugation device (Nanosep with Omega Membrane, Fisher Scientific, Schwerte, Germany). Samples were centrifuged at 9600× *g*, respectively, at room temperature until the volume passed the filter completely. This step was repeated once. Reduction and alkylation were performed by adding 100 μL dithiothreitol (DTT) (50 mM DTT in UA buffer) to the filter device and incubating for 5 min at 60 °C and 600 rpm (MultiTherm shaker, Benchmark Scientific, NJ, Sayreville, USA). This step was followed by centrifugation at 9600× *g* at room temperature (until the volume passed the filter completely). Next, 100 µL of iodoacetamide (IAA) buffer (50 mM IAA in UA buffer) were added to the filter device, followed by incubation for 1 min at room temperature with 600 rpm and centrifugation (9600× *g* at room temperature until volume passed the filter completely). The filter device was washed three times by adding 150 μL of UA buffer. A tryptic digest (trypsin/LysC, Mass Spec Grade, Promega, Mannheim, Germany) with 10 ng/μL in DB buffer (50 mM Tris-HCl, pH 8.5, 50 µL) was performed overnight at 37 °C and 300 rpm shaking. On the next day, samples were centrifuged (9600× *g* at room temperature until volume passed the filter completely), and peptide-containing flowthrough was analyzed by absorption measurement (205 nm, NanoDrop, Thermo Fisher Scientific, Darmstadt, Germany) for peptide content. From each sample, 20 µg peptide was dried in a SpeedVac and then dissolved in 10 μL 2.5% *v/v* acetonitrile (ACN)/0.1% trifluoroacetic acid (TFA) in LC-MS grade water (Merck, Darmstadt, Germany). To remove particles, samples were centrifuged for 5 min at 13,000× *g*. Supernatant was used for Nano-Liquid Chromatography–Orbitrap Mass Spectrometry (nLC-Orbitrap MS/MS, Thermo Fisher Scientific, Dreieich, Germany) measurement.

### 4.6. Nano-Liquid Chromatography–Orbitrap Mass Spectrometry Measurement

In this study, 1 µg of peptide sample was loaded to the nano-HPLC system (UltiMate 3000 RSLC, Thermo Fisher, Dreieich, Germany) by autosampler. Desalting was performed with a C18 pre-column cartridge (Acclaim PepMap™ 100, 300 µm I.D. × 5 mm) and separated on a 25 cm C18 column (Acclaim™ PepMap™ 100, 2 µm particle size, 75 µm I.D. × 25 cm; all parts were purchased from Thermo Fisher Scientific, Dreieich, Germany). An effective gradient of 5–30% of solvent B (80% ACN, 0.1% formic acid (FA) in LC-MS grade water mixed with solvent A (0.1% FA in LC-MS grade water) in 120 min with a flow rate of 300 nL/min was performed. Online ESI-Orbitrap mass spectrometry measurements were carried out by a Q Exactive Plus instrument (Thermo Fisher Scientific, Dreieich, Germany), equipped with a dual MALDI-/ESI-source (Spectroglyph LLC, Kennewick, WA, USA) in data-dependent top 10 acquisition mode. The MS scan range was 350–2000 m/z with a resolution of 70,000, and the dynamic exclusion time of precursors for MS/MS was set to 60 s. Precursor ions were scanned with a resolution of 17,500 and fragmented with a normalized collision energy (CE) of 28.

### 4.7. Data Analysis: Protein Identification and Label-Free Quantification

Protein identification and quantification were performed with Proteome Discoverer versions 2.4 and 3.0 with precursor detection node (Thermo Fisher Scientific). The human protein database (Homo sapiens SwissProt DB (TaxID 9606), Version 25 October 2017) was used as a template for peptide spectrum matching. For the protein identification and label-free quantification (LFQ), mass tolerances of 10 ppm for precursor ions and 0.02 Da for deviation from the theoretical masses from MS/MS were allowed. The maximum number of missed cleavages for tryptic digest was set to two. In the validation steps, a false discovery rate (FDR) of 0.01 was selected for protein and peptide identification. Dynamic modifications of oxidation (M) and deamidation (N, Q) and fixed modification of carbamidomethylation (C) were set. LFQ values were normalized based on total peptide amount. Protein abundance was calculated based on the average of the top-three quantified peptides (top N set to 3) of normalized LFQ values. Fold changes (abundance ratios) were calculated based on the protein abundance setting larger values to a maximum of 100 fold change. Statistical quantitative analysis and principal component analysis (PCA) were performed with Proteome Discoverer software. The analysis of top regulated proteins (Section 4.2) is based on the evaluation with version PD 2.4 (according to data in Appendix A_PDv24), and all other analyses were performed with PD 3.0 (according to data in Appendix A_PDv30). The mass spectrometry proteomics raw data were deposited to the ProteomeXchange Consortium via the PRIDE [118] partner repository with the dataset identifier PXD062079.

For protein identification, only proteins with at least one unique peptide were allowed. For gene ontology enrichment analysis, the Search Tool of the Retrieval of Interacting Genes/Proteins (STRING, http://string-db.org/) was used for significantly regulated proteins. The following settings were used for analysis: Network: full STRING network; required score: medium confidence (0.4); FDR stringency: medium (5%); and minimum count in network: 2. Venn diagrams were generated with the online tool VENNY 2.1 (https://bioinfogp.cnb.csic.es/tools/venny/).

For analysis of top regulated proteins, we considered proteins as significantly regulated only if they held at least two unique peptides, a minimum fold change of 2 of abundance ratios (LFQ values), and a maximum *p*-value of 0.05. Volcano plots were generated with Python (Python.org, Python 3.12.4) based on log2-transformed fold changes and log10-transformed *p*-values generated by a two-sided *t*-test with significance values.

## 5. Conclusions

In this study, we compared untreated, mock-transfected, and, for rAAV production, transfected HEK-293 cells. Analyses of these data revealed interesting proteins, many of which are reported to play a role in the cell cycle, gene expression, and host cell response to viruses, or to be associated with adenoviral proteins. In the transfection process, three plasmids were added, expressing several proteins known to interfere with the cell cycle as well as several proteins resulting in the viral product. These complex effects on the nuclear host cell proteome are difficult to reconcile with one typical GO term or pathway. Our data underline the important roles of already published factors but also provide novel proteins in this context. These are promising targets for further investigation of their impact on rAAV production and on transfection in general. Depending on their regulation and known functions, approaches with small-molecule inhibitors, knockdown, knockout, or overexpression can be interesting to improve rAAV production. In this study, we utilized the GOI plasmid with an unrelated plasmid as a reference. Deciphering in more detail the isolated effect of the Helper or RepCap plasmid on the host cell by testing additional plasmid combinations would be of interest. Further on, an integrated look at the interplay of AAV production and host expression by including other transfection methods, further selected HEK-293 clones, RNAseq data, metabolic analyses, and proteomic studies will provide a more holistic view. Next to the nuclear fraction, these studies could include the cytoplasmic fraction and even specific organelles and the culture medium to gain a broader overview of regulated pathways and help to understand and optimize cellular processes for a reliable, high-titer viral-particle generation.

## Figures and Tables

**Figure 1 ijms-26-06315-f001:**
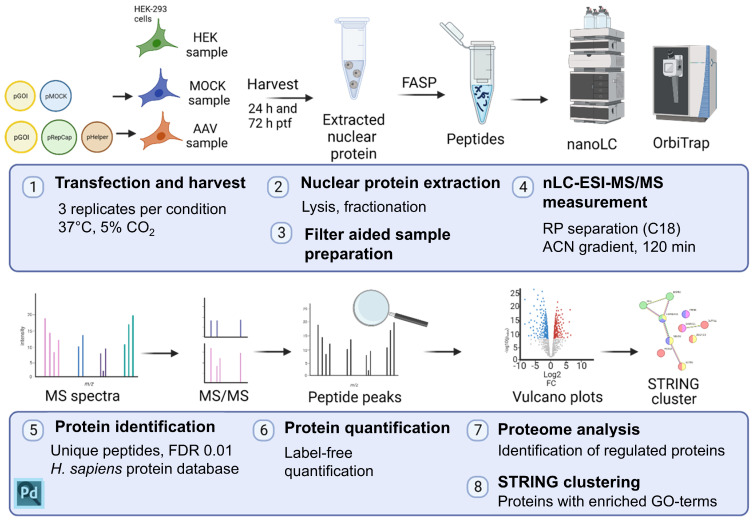
Schematic overview of the workflow of this study. (1) Untransfected (HEK), mock-transfected (MOCK), and rAAV-production-transfected (AAV) cells were harvested 24 h or 72 h after transfection (ptf) in biological triplicates. (2) Samples were prepared for MS/MS measurement by preparing nuclear protein fractions. (3) The nuclear proteins were further processed by the FASP method, including tryptic digest. (4) nLC-ESI-MS/MS measurement was performed with a 120 min gradient with acetonitrile (ACN) on a reverse phase (RP) separation (C18) column. (5, 6) Proteins were identified with Proteome Discoverer (PD) software and quantified in a label-free quantification approach. (7) Regulations of proteins were analyzed, and (8) enriched GO terms were investigated with STRING. Created with BioRender (https://biorender.com/).

**Figure 2 ijms-26-06315-f002:**
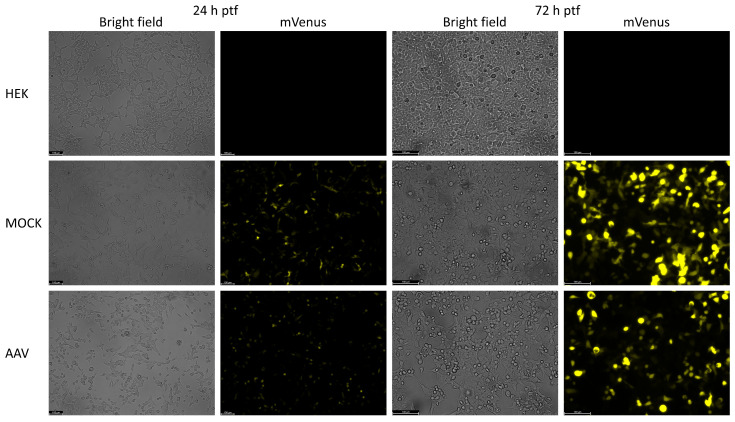
Bright field and fluorescence pictures of HEK-293 cells from HEK, MOCK, and AAV samples 24 h and 72 h post-transfection. After 24 h and 72 h, mVenus fluorescence was detected in MOCK and AAV samples, indicating successful transfection of the pITR-mVenus plasmid.

**Figure 3 ijms-26-06315-f003:**
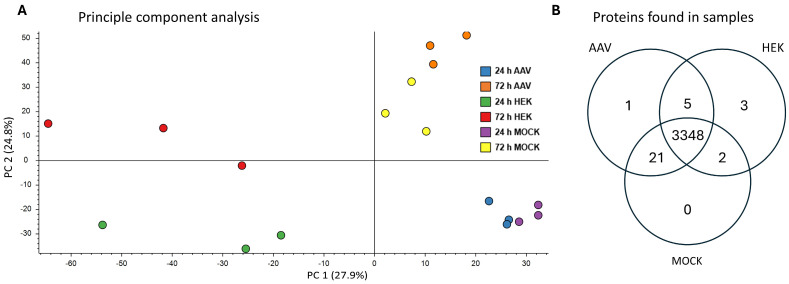
(**A**) Principal component analysis (PCA) plot of the biological triplicates of all six sample groups, generated with Proteome Discoverer (PD) software. In this figure, 24 h ptf HEK control samples are in green; rAAV-transfected are in blue; and MOCK-transfected are in purple. Further, 72 h ptf HEK controls are in red; rAAV-transfected are in orange; and MOCK-transfected are in yellow. (**B**) Distribution of identified proteins for AAV, HEK, and MOCK samples at both time points combined.

**Figure 4 ijms-26-06315-f004:**
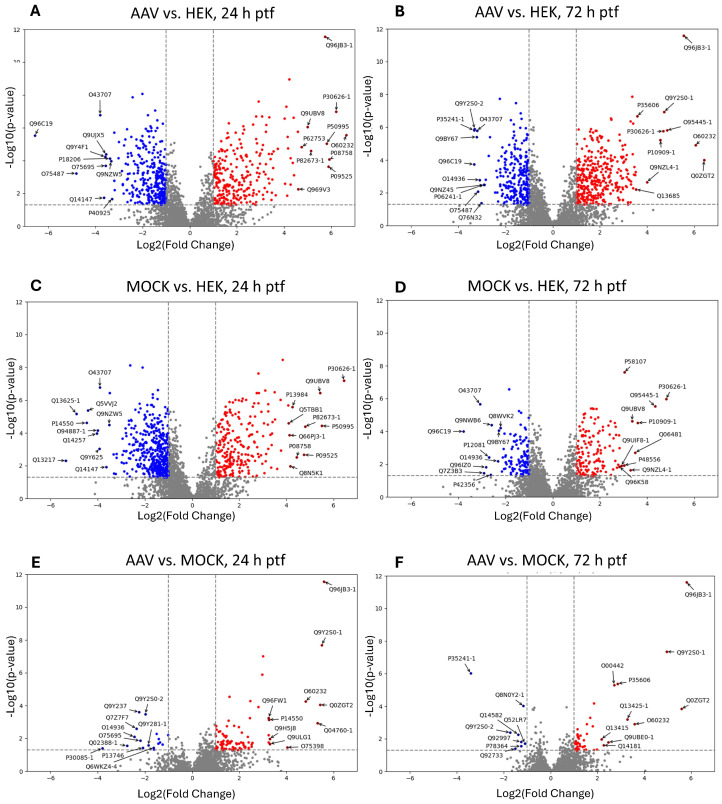
Analysis of significantly regulated proteins in transfected compared to untransfected HEK-293 cells. Samples were compared according to the time points of harvest, 24 h or 72 h post-transfection. Volcano plots were generated with a cut-off of <0.05 for the *p*-value and a log2(fold change) of >1, respectively, shown as dashed lines (blue color highlights significantly downregulated and red upregulated proteins). The 10 strongest regulated proteins (fold change) are labeled with UniProt accession numbers. Panels (**A**–**F**) show the comparisons given in the respective panel titles (AAV vs. HEK, MOCK vs. HEK, AAV vs. MOCK) with 24 h ptf on the left and 72 h ptf on the right side.

**Figure 5 ijms-26-06315-f005:**
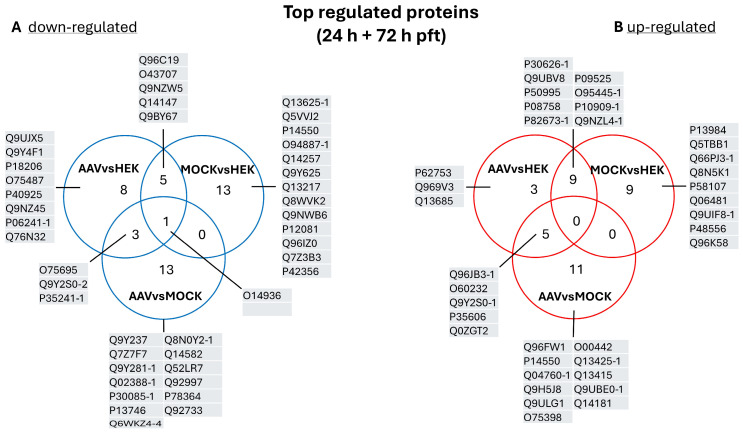
Venn diagram of top regulated proteins. The top ten regulated proteins were highlighted in the volcano plots (Figure 4), and the top regulated proteins of 24 h and 72 h ptf were merged for each comparison (AAV vs. HEK, MOCK vs. HEK, and AAV vs. MOCK). The number of regulated proteins can exceed ten when different proteins are regulated at different time points out of each comparison (see volcano plots, Figure 4). (**A**) Top down-regulated proteins and (**B**) up-regulated proteins. UniProt accessions numbers are listed for proteins of each group. A detailed list of detected proteins can be found in the Appendix A.

**Figure 6 ijms-26-06315-f006:**
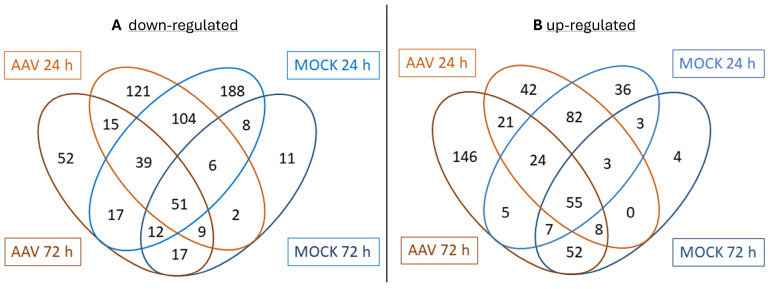
Venn diagram of (**A**) significantly down-regulated or (**B**) up-regulated proteins in each sample group (versus HEK-untransfected) and their overlaps.

**Figure 7 ijms-26-06315-f007:**
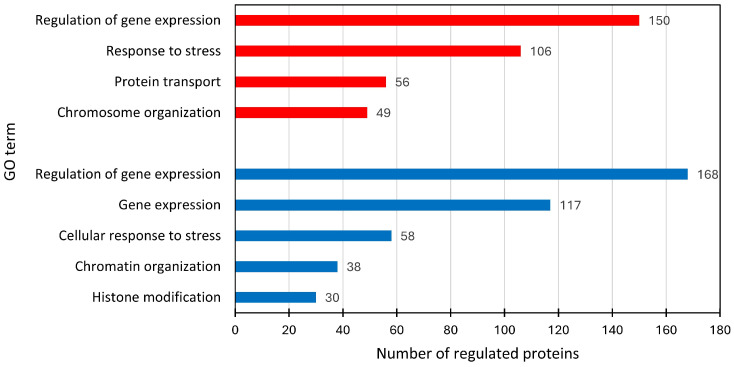
GO terms of down-regulated (blue) and up-regulated (red) proteins in AAV samples, identified by STRING analysis.

**Table 1 ijms-26-06315-t001:** Top down-regulated proteins. Proteins are listed with name, UniProt accession number (sorted in ascending order per group), and fold changes at 24 h and 72 h ptf. At each time point, fold changes in significant top regulated proteins are displayed in bold.

Group	Protein	UniProt	Fold Change					
			AAV/HEK		MOCK/HEK		AAV/MOCK	
			24 h	72 h	24 h	72 h	24 h	72 h
**AAV vs. HEK**	Glypican-4	O75487	**0.04 ***	**0.11 ***	0.13 *	0.19	0.26	0.59
	Tyrosine-protein kinase Fyn	P06241-1	0.11 *	**0.13 ***	0.13 *	0.29	0.84	0.46
	Vinculin	P18206	**0.10 ***	0.39 *	0.25 *	0.45	0.39	0.87
	Malate dehydrogenase, cytoplasmic	P40925	**0.10 ***	0.27	0.19	0.43	0.54	0.61
	Centrosomal protein of 68 kDa	Q76N32	3.17	**0.12 ***	0.53	0.28	5.96	0.44
	CDGSH iron-sulfur domain-containing protein 1	Q9NZ45	0.69	**0.13 ***	0.71	0.22	0.97	0.59
	Anaphase-promoting complex subunit 4	Q9UJX5	**0.09 ***	0.58	0.14 *	0.41	0.6	1.44
	FERM, RhoGEF, and pleckstrin domain-containing protein 1	Q9Y4F1	**0.08 ***	0.39	0.22 *	0.39	0.38	1.01
**MOCK vs. HEK**	FERM, RhoGEF, and pleckstrin domain-containing protein 2	O94887-1	0.21	0.42	**0.06 ***	0.24 *	3.43	1.78
	Histidine-tRNA ligase	P12081	0.43	0.35	0.42	**0.15 ***	1.01	2.26
	Alcohol dehydrogenase [NADP(+)]	P14550	0.43	1.41	**0.05 ***	1.07	9.56	1.32
	Phosphatidylinositol 4-kinase alpha	P42356	1.3	0.57	0.29	**0.16 ***	4.58	3.54
	DnaJ homolog subfamily C member 3	Q13217	0.17	1.75	**0.03 ***	0.96	6.74	1.83
	Apoptosis-stimulating of p53 protein 2	Q13625-1	0.17 *	0.36	**0.03 ***	0.53	5.09	0.68
	Reticulocalbin-2	Q14257	0.16 *	0.88	**0.06 ***	0.25 *	2.6	3.54
	Histone H2A deubiquitinase MYSM1	Q5VVJ2	0.06	0.26	**0.05 ***	0.43	1.29	0.61
	KAT8 regulatory NSL complex subunit 1	Q7Z3B3	0.24	0.27	0.14 *	**0.13 ***	1.77	2.04
	U4/U6.U5 small nuclear ribonucleoprotein 27 kDa protein	Q8WVK2	0.7	0.15 *	0.43 *	**0.22 ***	1.63	0.7
	PRKC apoptosis WT1 regulator protein	Q96IZ0	1.34	0.38	0.86	**0.14 ***	1.55	2.7
	Arginine- and glutamate-rich protein 1	Q9NWB6	0.54	0.24 *	0.50	**0.17 ***	1.07	1.45
	Glypican-6	Q9Y625	0.14 *	0.4	**0.07 ***	0.57	2.11	0.7
**AAV vs. MOCK**	HLA class I histocompatibility antigen, A-11 alpha chain	P13746	0.35	1.25	1.46	1.46	**0.24 ***	0.85
	UMP-CMP kinase	P30085-1	0.29	1.49	4.06	0.67	**0.07 ***	2.24
	Polyhomeotic-like protein 1	P78364	0.30 *	0.26 *	0.57	0.63	0.52	**0.42 ***
	Collagen alpha-1(VII) chain	Q02388-1	0.31	0.9	2.12	1.38	**0.15 ***	0.65
	Max dimerization protein 4	Q14582	0.12 *	0.25 *	0.35 *	0.65	0.35 *****	**0.38 ***
	Enhancer of polycomb homolog 2	Q52LR7	0.29 *	0.22 *	0.34 *	0.47	0.85	**0.47 ***
	Rab11 family-interacting protein 1	Q6WKZ4-4	0.36	1.94	1.13	1.94	**0.32 ***	1.00
	39S ribosomal protein L55, mitochondrial	Q7Z7F7	2.03	2.69	10.41 *	1.77	**0.20 ***	1.52
	Zinc finger protein 444	Q8N0Y2-1	0.76	0.45 *	1.23	1	0.62	**0.45 ***
	Proline-rich protein PRCC	Q92733	0.25 *	0.19 *	0.27 *	0.54	0.93	**0.35 ***
	segment polarity protein disheveled homolog DVL-3	Q92997	2.61	0.69	4.32 *	1.66	0.60	**0.42 ***
	Peptidyl-prolyl cis-trans isomerase NIMA-interacting 4	Q9Y237	0.15 *	0.82	0.72	0.8	**0.21 ***	1.02
	Cofilin-2	Q9Y281-1	0.67	0.81	2.44	1.21	**0.28 ***	0.67
**AAV vs. HEK and MOCK vs. HEK**	Alpha-actinin-4	O43707	**0.07 ***	**0.11 ***	**0.07 ***	**0.12 ***	1.08	0.92
	Probable ATP-dependent RNA helicase DHX34	Q14147	**0.08 ***	0.07	**0.08 ***	0.13	1	0.52
	EF-hand domain-containing protein D2	Q96C19	**0.01 ***	**0.10 ***	0.01	**0.07 ***	1.19	1.35
	Cell adhesion molecule 1	Q9BY67	0.14 *	**0.11 ***	0.16 *	**0.21 ***	0.92	0.52
	MAGUK p55 subfamily member 6	Q9NZW5	**0.10 ***	0.21 *	**0.09 ***	0.25 *	1.13	0.86
**AAV vs. HEK and AAV vs. MOCK**	Protein XRP2	O75695	**0.09 ***	0.20 *	0.39	0.43	**0.22 ***	0.47
	Radixin	P35241-1	1.13	**0.10 ***	1.13	1.05	1.00	**0.09 ***
	Isoform 2 of DNA-directed RNA polymerases I and III subunit RPAC2	Q9Y2S0-2 P0DPB5	0.25 *	**0.10 ***	0.98	0.33 *	**0.25 ***	**0.30 ***
**AAV vs. MOCK and MOCK vs. HEK and AAV vs. HEK**	Peripheral plasma membrane protein CASK	O14936	0.14 *	**0.12 ***	0.76	**0.20 ***	**0.18 ***	0.59

* significant fold change based on a *t*-test with a threshold of 0.05.

**Table 2 ijms-26-06315-t002:** Top up-regulated proteins. Proteins are listed with name, UniProt accession number (sorted in ascending order per group), and fold changes at 24 h and 72 h ptf. At each time point, fold changes in significant top regulated proteins are displayed in bold.

Group	Protein	UniProt	Fold Change					
			AAV/HEK		MOCK/HEK		AAV/MOCK	
			24 h	72 h	24 h	72 h	24 h	72 h
**AAV vs. HEK**	40S ribosomal protein S6	P62753	**26.9 ***	4.1 *	5.98 *	3.44	4.5	1.18
	Angio-associated migratory cell protein	Q13685	1.5	**11.6 ***	0.91	4.31	1.69	2.7
	Nicalin, BOS complex subunit NCLN	Q969V3	**24.2 ***	1.8	6.48	1.66	3.73	1.11
**MOCK vs. HEK**	General transcription factor IIF subunit 2	P13984	23.93 *	8.74 *	**19.1 ***	6.5 *	1.26	1.35
	26S proteasome non-ATPase regulatory subunit 8	P48556	1.11	11.46 *	1.1	**8.1 ***	0.98	1.41
	Epiplakin	P58107	18.55 *	10.31 *	14.4 *	**8.2 ***	1.29	1.26
	Amyloid-like protein 2	Q06481	2.29	8.78 *	5.2 *	**11.3 ***	0.44	0.78
	Ribonuclease h2 subunit b	Q5TBB1	17.89 *	10.44 *	**17.2 ***	7.0 *	1.04	1.5
	ADP-ribosylation factor-like protein 6-interacting protein 4	Q66PJ3-1	19.43 *	1.14	**17.5 ***	1.2	1.11	0.94
	CDGSH iron-sulfur domain-containing protein 2	Q8N5K1	17.4	1.19	**17.8 ***	0.7	0.97	1.63
	Zinc finger protein 668	Q96K58	1.3	4.21	1.8	**7.1 ***	0.71	0.59
	Bromodomain adjacent to zinc finger domain protein 2B	Q9UIF8-1	12.49	10.62 *	12.2 *	**7.5 ***	1.03	1.42
**AAV vs. MOCK**	RNA 3′-terminal phosphate cyclase	O00442	0.01	2.84 *	0.26	0.43 *	0.01	**6.5 ***
	Deformed epidermal autoregulatory factor 1 homolog	O75398	3.46	0.64	0.21	0.57	**16.3 ***	1.1
	Alcohol dehydrogenase [NADP(+)]	P14550	0.43	1.41	0.04 *	1.07	**9.6 ***	1.3
	Lactoylglutathione lyase	Q04760-1	4.09	1.02	0.1	0.58	**40.0 ***	1.8
	Origin recognition complex subunit 1	Q13415	9.73	1.52	2.33	0.34	4.2	**4.5 ***
	Beta-2-syntrophin	Q13425-1	0.35	3.86	0.51	0.4	0.7	**9.6 ***
	DNA polymerase alpha subunit B	Q14181	0.31	3.05	0.01	0.64	≥100	**4.8 ***
	Ubiquitin thioesterase otub1	Q96FW1	3.9	2.13	0.41	0.66	**9.5 ***	3.2
	TATA box-binding protein-associated factor RNA polymerase I subunit D	Q9H5J8	19.23 *	0.78	1.97	0.74	**9.8 ***	1.0
	SUMO-activating enzyme subunit 1	Q9UBE0-1	1.07	2.78	0.34	0.5	3.2	**5.5 ***
	DNA helicase ino80	Q9ULG1	4.21	1.38	0.43	0.96	**9.9 ***	1.4
**AAV vs. HEK and MOCK vs. HEK**	Apolipoprotein M	O95445-1	4.2 *	**28.8 ***	4.2 *	**20.3 ***	1.02	1.42
	Annexin A5	P08758	**60.0 ***	4.4	**21.6 ***	6.4	2.77	0.68
	Annexin A4	P09525	**59.1 ***	4.8	**26.8 ***	3.6	2.2	1.33
	Clusterin	P10909-1	≥100	**23.7 ***	≥100	**12.2 ***	1.78	1.95
	Sorcin	P30626-1	**74.1 ***	**25.9 ***	**86.7 ***	**28.1 ***	0.85	0.92
	Annexin A11	P50995	**55.9 ***	6.7	**45.2 ***	6.2	1.23	1.08
	28S ribosomal protein S35, mitochondrial	P82673-1	**35.4 ***	3.3	**27.7 ***	2.9	1.27	1.14
	Hsp70-binding protein 1	Q9NZL4-1	1.7	**16.0 ***	1.9	**9.8 ***	0.93	1.63
	Peflin	Q9UBV8	**31.8 ***	7.5 *	**43.2 ***	**10.3 ***	0.73	0.72
**AAV vs. HEK and AAV vs. MOCK**	ZNRD2	O60232	**≥100 ***	**66.7 ***	3.54	5.54	**28.3 ***	**12.0 ***
	Coatomer subunit beta	P35606	17.1 *	**12.0 ***	2.19	1.64	7.8 *	**7.3 ***
	Nexilin	Q0ZGT2	21.3 *	**85.8 ***	0.49	1.77	**43.2 ***	**48.3 ***
	Hypermethylated in cancer 2 protein	Q96JB3-1	**53.3 ***	**47.2 ***	1.11	0.84	**48.1 ***	**55.9 ***
	DNA-directed RNA polymerases I and III subunit RPAC2	Q9Y2S0-1 P0DPB6	21.1 *	**26.5 ***	0.467	0.851	**45.2 ***	**31.1 ***

* significant fold change based on a *t*-test with a threshold of 0.05.

**Table 3 ijms-26-06315-t003:** List of regulated proteins assigned to the GO term ‘response to virus’ with their UniProt accession number, name, abbreviation, gene symbol, and fold changes. Fold changes are colored by range: white < 2, light pink < 4, pink < 8, and red ≥ 8.

UniProt Accession	Description	Protein	Gene Symbol	Fold Change AAV vs. HEK		Fold Change MOCK vs. HEK		Fold Change AAV vs. MOCK	
				24 h	72 h	24 h	72 h	24 h	72 h
P10909-1	Clusterin	CLUS	CLU	4.9 *	20.9 **	4.2	9.2 **	1.2	2.3
Q9Y3Z3	Deoxynucleoside triphosphate triphosphohydrolase SAMHD1	SAMH1	SAMHD1	4.2 *	10.8 **	1.0	4.0 *	4.2 *	2.7
Q7Z7L1	Schlafen family member 11	SLN11	SLFN11	8.1 *	6.2 **	4.3 *	2.5	1.9	2.5
Q9BYW2	Histone-lysine N-methyltransferase SETD2	SETD2	SETD2	1.4	4.3 **	0.6	1.5	2.2	2.9
Q6L8Q7-1	2′,5′-phosphodiesterase 12	PDE12	PDE12	n/a ***	3.2 *	n/a	1.3	n/a	2.4 *
O00571	ATP-dependent RNA helicase DDX3X	DDX3X	DDX3X	2.7 **	2.6 **	2.0 **	2.2 **	1.3	1.2
P04792	Heat shock protein beta-1	HSPB1	HSPB1	0.9	2.5 *	1.9	9.9 *	0.9	1.2
Q9NUD5	Zinc finger CCHC domain-containing protein 3	ZCHC3	ZCCHC3	4.8 **	2.3 *	5.9 **	1.8	0.8	1.2
Q9C030	Tripartite motif-containing protein 6	TRIM6	TRIM6	0.7	2.1 *	0.8	1.8	0.9	1.2
P07900	Heat shock protein HSP 90-alpha	HSP90AA1	HSP90AA1	0.6	2.0 **	0.6	1.2	1.1	1.7 *

* significant fold change calculated in a *t*-test with a threshold of 0.05; ** significant fold changes, based on Benjamini-corrected *p*-values of the *t*-test with a threshold of 0.05; *** ‘n/a’ abbreviates ‘not available’.

**Table 4 ijms-26-06315-t004:** Regulated proteins with context to antiviral response in HEK-293 cells producing rAAV upon plasmid transfection identified by literature search. Proteins are listed with their UniProt accession number, name, abbreviation, gene symbol, and fold changes. Fold changes are colored by range: white < 2, light pink < 4, pink < 8.

UniProt Accession	Description	Protein	Gene Symbol	Fold Change AAV vs. HEK		Fold Change MOCK vs. HEK		Fold Change AAV vs. MOCK	
				24 h	72 h	24 h	72 h	24 h	72 h
**O00743**	Serine/threonine-protein phosphatase 6 catalytic subunit	PPP6C	PPP6C	1.1	2.8 *	0.8	1.1	1.3	2.6
**Q9NUD5**	Zinc finger CCHC domain-containing protein 3	ZCHC3	ZCCHC3	4.8 **	2.3 *	5.9 **	1.8	0.8	1.2

* significant fold change calculated in a *t*-test with a threshold of 0.05; ** significant fold changes, based on Benjamini-corrected *p*-values of the *t*-test with a threshold of 0.05.

**Table 5 ijms-26-06315-t005:** Regulated proteins with context to adenoviral proteins identified by literature search. Proteins are listed with UniProt accession number, name, abbreviation, gene symbol, and fold changes. Fold changes are colored by range: deep blue ≤ 0.2, blue ≤ 0.5, white < 2, light pink < 4, pink < 8.

UniProt Accession	Description	Protein	Gene Symbol	Fold Change AAV vs. HEK		Fold Change MOCK vs. HEK		Fold Change AAV vs. MOCK	
				24 h	72 h	24 h	72 h	24 h	72 h
P63151	Serine/threonine-protein phosphatase 2A 55 kDa regulatory subunit B alpha isoform	2ABA	PPP2R2A	1.3	2.6 *	0.6	1.3	2.2 *	2
P30154	Serine/threonine-protein phosphatase 2A 65 kDa regulatory subunit A beta isoform	2AAB	PPP2R1B	0.7	0.6	0.3 **	0.6	2.3 *	0.9
P01106	Myc proto-oncogene protein	MYC	MYC	0.6	0.7	0.7	0.9	0.9	0.8
P26196	Probable ATP-dependent RNA helicase DDX6	DDX6	DDX6	2.1 **	1.9 **	1.6 *	1.5	1.3	1.3
P04637	Cellular tumor antigen p53	P53	TP53	0.8	0.6 **	0.9	0.8	0.9	0.7
Q92878	DNA repair protein Rad50	RAD50	RAD50	0.4 *	0.2 **	0.2	0.6	1.7	0.3 *
Q93034	Cullin-5	CUL5	CUL5	2.7 *	5.4 **	1.6	2	1.7	2.7

* significant fold change calculated in a *t*-test with a threshold of 0.05; ** significant fold changes, based on Benjamini-corrected *p*-values of the *t*-test with a threshold of 0.05.

## Data Availability

The mass spectrometry proteomics raw data were deposited to the ProteomeXchange Consortium via the PRIDE partner repository with the dataset identifier PXD062079.

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
