# Peer review of "Nuclear Fraction Proteome Analyses During rAAV Production of AAV2-Plasmid-Transfected HEK-293 Cells"

_ijms, 2025, doi:10.3390/ijms26136315_

Round 1

Reviewer 1 Report

Comments and Suggestions for Authors

In this manuscript, Golm et al are exploring the proteomic profile of HEK-293 nuclear fractions after transfection with an AAV ITR plasmid alone compared to triple transfected HEK-293s that produce rAAV.  Overall, the manuscript is well-written, and the topic is of interest to the AAV gene therapy community.  However, there are several concerns that need to be addressed:

  • The authors state that other groups have performed proteomic profiling experiments similar to this; however, they chose to focus on the nuclear fraction. Although rAAV assembly occurs in the nucleus, the approach to examine only the nuclear fraction limits the interpretation of the data as it cannot determine the contribution of proteins that are exported/imported in response to viral production.
  • The authors chose to compare an untransfected HEK-293 sample with an AAV ITR plasmid transfected sample, and a triple-transfected sample.  Why didn’t the authors compare the individual samples for the Ad-Helper plasmid and the Rep/Cap expressing plasmid as well?  This would have provided a better understanding of which proteins were truly affected by viral production, rather than just overexpression of certain viral proteins in a plasmid context.
  • Figure 1 appears in duplicate.
  • The setup of Tables 1-9 is a little clunky.  Authors should consider one single table with all samples (Untransfected, MOCK, and AAV) with headers above sections that group the proteins by function.
  • The discussion reads as if all affected proteins were just listed and then a few sentences regarding reported functions of that protein are mentioned. It would be more helpful to the reader if the discussion were reorganized into larger sections and the relevance of each section to rAAV production highlighted.  For example, instead of organizing by listing proteins that are upregulated/downregulated, it would be much more conceptually useful to discuss proteins involved in DNA repair/cell cycle pathways together with some being downregulated and some upregulated to enhance rAAV production.  Furthermore, the known functions of each protein should be defined regarding its potential relevance to rAAV production, rather that just listing that the protein is a deubiquitinase.

Author Response

In this manuscript, Golm et al are exploring the proteomic profile of HEK-293 nuclear fractions after transfection with an AAV ITR plasmid alone compared to triple transfected HEK-293s that produce rAAV.  Overall, the manuscript is well-written, and the topic is of interest to the AAV gene therapy community.  However, there are several concerns that need to be addressed:

  • We thank the reviewer to judge our manuscript to be well written and being on a topic of interest to the AAV community

The authors state that other groups have performed proteomic profiling experiments similar to this; however, they chose to focus on the nuclear fraction. Although rAAV assembly occurs in the nucleus, the approach to examine only the nuclear fraction limits the interpretation of the data as it cannot determine the contribution of proteins that are exported/imported in response to viral production.

  • As stated in our introduction we purposely focused on the nucleus as this is the compartment of gene expression, viral assembly and GOI packaging. If the nuclear protein abundance would change due to import or export, we would detect the change. We agree, that further studies of more cellular fractions and even the medium are warranted and might give a broader picture. We added a sentence to our conclusion to provide the respective outlook.

The authors chose to compare an untransfected HEK-293 sample with an AAV ITR plasmid transfected sample, and a triple-transfected sample.  Why didn’t the authors compare the individual samples for the Ad-Helper plasmid and the Rep/Cap expressing plasmid as well?  This would have provided a better understanding of which proteins were truly affected by viral production, rather than just overexpression of certain viral proteins in a plasmid context.

  • Our samples included (i) HEK: HEK-293 untransfected, (ii) AAV: HEK-293 triple transfected for AAV production and (iii) MOCK: HEK-293 transfected with the ITR/GOI Plasmid and a pUC plasmid. By comparing the difference of AAV sample versus the MOCK sample we expect to capture contributions of the non-backbone pRepCap and pHelper effects. As a technical bonus aspect, our GOI plasmid enables a transfection control, which ensures comparability between the samples. We agree that testing all 2-Plasmid and 3-Plasmid combinatorial options of the four plasmids pGOI, pRepCap, pHelper, pUC would provide a broader picture. However, conceptually 2 out of 4 yields 6 and 3 out of 4 a further 4 combinations. In addition, different transfection methods may also influence the outcome. We posit that our data already provide the most crucial information and further combinations can be studied in the future. As these are important considerations, we added the reasoning to our conclusion.  

Figure 1 appears in duplicate.

  • We apologize for this mistake and fixed it (there was a hidden code in the text, which regenerated the figure upon a cross-reference update).

The setup of Tables 1-9 is a little clunky.  Authors should consider one single table with all samples (Untransfected, MOCK, and AAV) with headers above sections that group the proteins by function.

  • We optimized the layout of the tables by combining the first six tables in two tables containing either the up- or down- regulated proteins. We tried to implement your suggestion of grouping the proteins by function but, unfortunately, we did not find overarching similarities that would justify such a grouping.

The discussion reads as if all affected proteins were just listed and then a few sentences regarding reported functions of that protein are mentioned. It would be more helpful to the reader if the discussion were reorganized into larger sections and the relevance of each section to rAAV production highlighted.  For example, instead of organizing by listing proteins that are upregulated/downregulated, it would be much more conceptually useful to discuss proteins involved in DNA repair/cell cycle pathways together with some being downregulated and some upregulated to enhance rAAV production.  Furthermore, the known functions of each protein should be defined regarding its potential relevance to rAAV production, rather that just listing that the protein is a deubiquitinase.

  • We tried to implement your suggestion but, also here, grouping and structuring the regulated proteins based on their impact on rAAV production was elusive. More studies are required to create a reliable background in this context. Therefore, our approach is to list first the most regulated proteins and secondly provide an additional evaluation of our results with proteins reported to be involved in response to virus or adeno viral proteins or which have already been published as related to rAAV, even if the proteins in latter three groups are not within the top regulated proteins. Positively speaking, the AAV agnostic nature of proteome data provides a broader view on the wide regulations observed during AAV production. More research is need to connect the dots and, in the long run, a new GO term tentatively named “AAV production related” may emerge. In this sense, the currently incoherent list of proteins provides one pillar to understand, construct and improve an AAV production pathway.

Reviewer 2 Report

Comments and Suggestions for Authors

This manuscript presents a well-executed and timely study on the nuclear proteome dynamics of HEK-293 cells during rAAV2 production. The authors performed comprehensive MS-based proteomic profiling and provided a large dataset of nuclear protein changes under different transfection conditions. The work is technically sound and addresses a relevant question in the field of viral vector production. The manuscript fits well within the scope of IJMS. While the overall structure and findings are solid, a few points would benefit from clarification or refinement to improve the manuscript’s clarity and accessibility:

1.Figures 3–6 (particularly volcano plots and Venn diagrams) are informative but could benefit from enhanced resolution and slightly larger font size for better readability.

2.Tables 1–6 are comprehensive but lengthy. Consider summarizing key regulated proteins in the main text and moving full versions to Supplementary Materials.

3.A summary heatmap or clustering visualization (e.g., top 50 changing proteins over time) could help the reader grasp temporal trends.

4.Ensure all abbreviations (e.g., MOCK, ITR, ptf) are defined clearly upon first use.

5.It may be helpful to include a visual schematic of the experimental design to guide the reader.

6.The GO term analysis results are interesting; consider including representative pathway or network diagrams (e.g., STRING network visualization) for enhanced impact.

7.A general language polishing would enhance clarity, particularly in the Results and Discussion sections where dense data interpretation is presented.

8.There are a few long or compound sentences (e.g., in Section 4.2) that could be broken down for better comprehension.

This study makes a valuable contribution to the field of rAAV production and nuclear proteome research. With minor revisions to improve clarity and data presentation, the manuscript will be suitable for publication in IJMS.

Author Response

This manuscript presents a well-executed and timely study on the nuclear proteome dynamics of HEK-293 cells during rAAV2 production. The authors performed comprehensive MS-based proteomic profiling and provided a large dataset of nuclear protein changes under different transfection conditions. The work is technically sound and addresses a relevant question in the field of viral vector production. The manuscript fits well within the scope of IJMS. While the overall structure and findings are solid, a few points would benefit from clarification or refinement to improve the manuscript’s clarity and accessibility:

1.Figures 3–6 (particularly volcano plots and Venn diagrams) are informative but could benefit from enhanced resolution and slightly larger font size for better readability.

  • We optimized the resolution and font size of that pictures.

2.Tables 1–6 are comprehensive but lengthy. Consider summarizing key regulated proteins in the main text and moving full versions to Supplementary Materials.

  • We summarized the up- and down-regulated proteins of tables 1-6 in two tables to make the data more accessible and provide a more stramlined presentation.

3.A summary heatmap or clustering visualization (e.g., top 50 changing proteins over time) could help the reader grasp temporal trends.

  • We tested coloring the two tables (up- and down-regulated proteins) but our impression was that this looked cluttered. Am additional, standalone heat map seemed to carry little information. Therefore, we did not change the appearance. We hope that the new, clarified table format serves alleviates your concerns.

4.Ensure all abbreviations (e.g., MOCK, ITR, ptf) are defined clearly upon first use.

  • Thank you for pointing this out to us. We checked all abbreviations.

5.It may be helpful to include a visual schematic of the experimental design to guide the reader.

  • We included a schematic overview as figure 1 to guide the reader through our study.

6.The GO term analysis results are interesting; consider including representative pathway or network diagrams (e.g., STRING network visualization) for enhanced impact.

  • The analysis only revealed minor pathway or network connections. You are right, we identified regulation of gene expression, response to stress, protein transport, chromosome organization and so on. As we were interested in the response of the host cell on rAAV production, we concentrated on the response to virus term. We could include this network from our STRING analysis but there are nearly no connections between these proteins. Therefore, we would prefer not to include this.

7.A general language polishing would enhance clarity, particularly in the Results and Discussion sections where dense data interpretation is presented.

  • We made small changes throughout the manuscript to the best of our language abilities.

8.There are a few long or compound sentences (e.g., in Section 4.2) that could be broken down for better comprehension.

  • We tested splitting sentences but this resulted in a lengthy text. Hence, we would prefer to keep it concise as some readers might want to just focus on proteins of their personal interest.

This study makes a valuable contribution to the field of rAAV production and nuclear proteome research. With minor revisions to improve clarity and data presentation, the manuscript will be suitable for publication in IJMS.